# A Hybrid Method for Vibration-Based Bridge Damage Detection

**Semih Gonen *** and **Emrah Erduran**

Department of Built Environment, Oslo Metropolitan University, 0166 Oslo, Norway
* Correspondence: semihgon@oslomet.no

**Abstract:** Damage detection algorithms employing the conventional acceleration measurements and the associated modal features may underperform due to the limited number of sensors used in the monitoring and the smoothing effect of spline functions used to increase the spatial resolution. The effectiveness of such algorithms could be increased if a more accurate estimate of mode shapes were provided. This study presents a hybrid structural health monitoring method for vibration-based damage detection of bridge-type structures. The proposed method is based on the fusion of data from conventional accelerometers and computer vision-based measurements. The most commonly used mode shape-based damage measures, namely, the mode shape curvature method, the modal strain energy method, and the modal flexibility method, are used for damage detection. The accuracy of these parameters used together with the conventional sparse sensor setups and the proposed hybrid approach is investigated in numerical case studies, with damage scenarios simulated on a simply-supported bridge. The simulations involve measuring the acceleration response of the bridge to ambient vibrations and train crossings and then processing the data to determine the modal frequencies and mode shapes. The efficiency and accuracy of the proposed hybrid health monitoring methodology are demonstrated in case studies involving scenarios in which conventional acceleration measurements fail to detect and locate damage. The robustness of the proposed method against various levels of noise is shown as well. In the studies considered, damage as small as 10% decrease in flexural stiffness of the bridge and localized in less than 1% of the span-length of the bridge is reliably detected even with very high levels of measurement noise. Finally, a modified modal flexibility damage parameter is derived and used to alleviate the shortcomings of the modal flexibility damage parameter.

**Keywords:** damage detection; vibration-based; structural health monitoring; computer vision; curvature; strain energy; modal flexibility

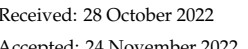

## 1. Introduction

Rapidly aging bridge infrastructure is becoming a major problem around the world. Most of the bridges in Europe and North America were built during the first half of the twentieth century and now have reached the end of their design life. For example, Europe's highway bridge count is around one million, and more than 35%, or half a million railway bridges, are over 100 years old, with many more close to the end of their 50-year design life [1]. Similarly, in the United States, more than 42% of the 617,000 bridges are at least 50 years old, 7.5% of the country's bridges are considered structurally deficient, and the estimated backlog cost of bridge repair is USD 125 billion [2]. An important increase in mobility, traffic, and urbanization, as well as climate change, is causing significant deviations from original design assumptions and accelerating the deterioration process [3].

Today, bridge maintenance decisions are mainly based on manual and visual inspections, and consume significant time and money. As such, detailed bridge inspections are generally performed at long intervals or after major events, leaving the infrastructure potentially vulnerable. Furthermore, visual inspections are strongly subjective, and the outcome of an inspection depends on the experience and skills of the inspector. This

prevents an objective comparison of the condition of different bridges. Structural Health Monitoring (SHM) techniques offer an alternative to manual inspection by generating objective real-time data on the condition of bridges. Of the different SHM techniques, vibration-based methods provide an attractive approach; the readily available vibrations of the structure can provide information about the behavior of the bridge at the global level and an opportunity for damage detection without prior knowledge of the damage location. Therefore, vibration-based damage detection in civil structures has been an intensively investigated subject in recent decades [4–17]. Numerous approaches have been developed that utilize the modal properties of structures or other properties originating from modal features [9,16,18,19]. Bridge structures have been the focus of several studies employing vibration signatures [1,7,20–23]. Long-term vibration analysis of bridges and the relationship between vibration and structural damage have been investigated in various studies [24–27]. The most common mode shape-based features used for damage detection and location include mode shape curvature, modal strain energy, and modal flexibility. For example, Dilena and Morassi [28] performed dynamic identification on a single-span bridge subjected to increasing levels of damage and used the mode shape curvature method to detect the location of the damage. Grande and Imbimbo [29] proposed a method based on modal flexibility to detect damage in multiple locations. Limongelli and Giordano [30] investigated the performance of these three damage parameters for damage localization and compared the results in terms of the information gain they provide. The same authors [31] tested these damage detection methods on a prestressed concrete bridge (the S101 Bridge) using the monitoring data. However, these methods are known to suffer significantly from measurement uncertainties and sensitivity of damage indicators to the location and severity of damage. One of the well-documented reasons for these shortcomings is the limited number of sensors commonly used in monitoring and the smoothing effect of spline functions fitted to discrete measurements to increase spatial resolution [32,33].

Computer vision provides an attractive alternative to alleviate this problem, as computer vision-based multipoint measurements can provide information with much higher spatial resolution at the critical regions of the structure. Further, low cost and non-contact measurements without the need for markers for accurate measurements significantly ease the application of computer vision-based measurements. Indeed, computer vision applications in civil engineering have gained significant momentum [34–36]. Consumer-grade and high-speed cameras have been used to extract modal frequencies and mode shapes under laboratory conditions [37–40]. Feng and Feng showed in both the laboratory [41] and field environment [42] that the accuracy of the measurements could be improved using markers in such situations where the camera is far away from the structure. Computer vision methods have been tested on real bridges [42–45] and other structures [46–48]. However, there are challenges with respect to the resolution and sensitivity of measured vibrations, as the size of the bridge, the field-of-view and resolution of the camera sensor, and the properties of the camera dictate the application. Considering that the deformations due to the vibrations of structures under service loads are very small compared to the length of the bridges, high-resolution sensors or several cameras are needed to capture the vibrations with high enough sensitivity to be able to detect subtle changes in the modal parameters due to damage. Further, monitoring of the middle sections of bridges with cameras can be challenging, as these sections are elevated and cross obstacles that make camera placement difficult.

In order to address these problems, the present study proposes a hybrid vibration-based structural health monitoring and damage detection methodology. The proposed methodology combines the use of conventional accelerometers with computer vision, thereby fusing the information from different sensors. This method leverages the increased spatial resolution of computer vision in the sensitive regions of the structure, providing virtually continuous information, while acceleration in other regions is measured using conventional sensors. More specifically, damage parameters based on changes in mode shapes have been well documented to be less reliable in support regions, while their performance is much better in the mid-span [32,33]. Therefore, increasing the spatial resolution

of the data collected near the abutments with computer vision while using conventional sensors in other parts of the bridge can lead to significantly improved damage detection and localization. As such, this method benefits from the strength of both approaches. As an example, in Figure 1, the computer vision system provides continuous vibration data for the highlighted portions, which are near the abutments of a bridge deck and are in the camera's area of sight, while conventional and sparsely distributed acceleration sensors measure the vibrations at discrete locations along the rest of the bridge deck.

In the proposed method, a standard video camera is used to measure the displacement and acceleration of a specific part of the bridge. The modal displacement in the part monitored using computer vision provides continuous modal information, which is combined with the modal displacement obtained from the conventional sparse measurement system data. As such, the amount of modal information about the bridge increases at critical locations and a better fit for a spline function can be achieved. Furthermore, the previously mentioned smoothing effect of the spline fit can be eliminated in the region measured using computer vision.

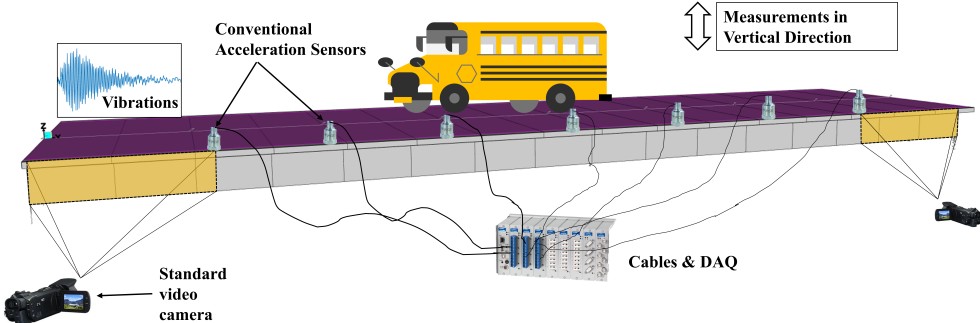

**Figure 1.** Proposed methodology illustrated on an isolated one-span bridge deck.

The proposed hybrid methodology [49] is tested using a numerical study to simulate (i) ideal conditions, (ii) high levels of excitation with train crossings over a bridge, and (iii) low levels of ambient vibration responses of a bridge combined with different levels of noise. A simply-supported bridge model is generated using the Finite Element (FE) method, then the acceleration signals obtained at the sensor locations are processed to determine the modal properties of the bridge. The benchmark case simulates a conventional vibration measurement system with low spatial resolution, whereas the hybrid monitoring system has increased resolution near the supports, where conventional approaches are known to struggle the most to detect and locate damage. A total of six case studies are conducted with various damage scenarios and acceleration levels to assess the efficacy of the proposed methodology.

## 2. Damage-Sensitive Features

This section briefly describes the vibration-based damage indicators that were developed in the early 1990s [33,50,51] and are widely used in vibration-based health monitoring. The common practice in vibration-based damage detection studies is to measure the vibration signals of the structure at discrete locations and employ the mode shapes and damage-sensitive features presented below. Dynamic identification using the measured signals provides modal information, such as the modal displacements at the measurement locations. Then, the spatial resolution of the mode shapes is increased by fitting continuous functions such as splines or polynomials to discrete measured values and interpolating the area between them to cover the entire space. This procedure is necessary in order to detect damage locations with higher accuracy, especially when the damage is in a narrow region that is further away from the sensor location. However, the function fitted to the discrete mode shape data often smooths the data, potentially concealing subtle changes in the mode shape due to damage [33].

In this study, we use three vibration-based damage indicators that are widely used in the structural health monitoring literature [52]. Other methods based on pattern recognition and novelty detection exist as well [53–55]; interested readers are referred to the reference studies in [19,33] for a detailed review of these methods.

### 2.1. Mode Shape Curvature

For structures that can be represented with the Bernoulli–Euler formulation under flexural actions, the curvature at a specific location $x$ can be calculated using the moment at that location, $M(x)$, along with the flexural stiffness $EI(x)$ of the beam cross-section. In Equation (1), the curvature $v''(x)$ is

$$v'' \approx \frac{M(x)}{EI} \tag{1}$$

where $E$ is the modulus of elasticity of the material and $I$ is the cross-sectional moment of inertia. Equation (1) shows that the curvature is inversely proportional to the flexural stiffness of the beam. Therefore, under the same loading conditions, damage in a given cross section (i.e., reduction in flexural stiffness) increases the curvature at the location of the damage. In this regard, changes in curvature can be tracked to detect and locate damage [50].

Mode shapes detected using Experimental or Operational Modal Analysis (EMA or OMA) techniques are discrete values at sensor locations. The spatial resolution of the detected mode shapes can be increased using continuous functions such as cubic spline in order to improve the estimation of the location of possible damage. Subsequently, the modal curvature of a beam at the discrete measurement points equally spaced at a distance $h$ can be approximated using the central difference theorem. Consequently, the second derivative of the modal displacement at the degree of freedom (DOF) $k$ of a mode shape $\Phi$ can be calculated using Equation (2):

$$v''(\Phi_{i,k}) \approx \frac{\Phi_{i,k-1} - 2\Phi_{i,k} + \Phi_{i,k+1}}{h^2} \tag{2}$$

where $v''(\Phi_{i,k})$ is the curvature of the $i$th mode shape at the $k$th DOF. The difference between the modal curvature of the possibly damaged state $d$ and the healthy or undamaged state $u$ is defined as the *Curvature Damage Index*, or CDI, and can be used to detect, locate, and eventually quantify any potential damage using the identified mode shapes

$$CDI_k = \sum_{i=1}^{nModes} |v''(\Phi_{i,k})^d - v''(\Phi_{i,k})^u|. \tag{3}$$

### 2.2. Modal Strain Energy

Another damage index based on modal features is the *modal strain energy*, which can be defined as the strain energy stored in a structure when it deforms purely in its mode-shape pattern, based on the idea that the distribution of the strain energy throughout the structure changes with damage. More specifically, when the stiffness of one segment of the structure is reduced due to sustained damage, it can no longer absorb the same amount of energy as when healthy [51]. This results in a deviation from the original strain energy distribution, which can then be used to detect and locate any potential damage.

If a Bernoulli–Euler beam is divided into $N$ subregions, then the energy stored in each subregion $j$ in the $i$th mode shape is provided by Equation (4a), while the total energy stored in the entire beam can be computed using Equation (4b).

$$U_{i,j} = \frac{1}{2} \int_{a_j}^{a_{j+1}} (EI)_j [v_i''(x)]^2 \, dx \tag{4a}$$

$$U_i = \frac{1}{2} \int_{0}^{L} EI [v_i''(x)]^2 \, dx \tag{4b}$$

where $a_j$ and $a_{j+1}$ are the start and end coordinates of subregion $j$ and $L$ is the length of the entire beam.

Assuming that the subregions are small enough and flexural rigidity is constant in the subregions, the fractional energy $F_{ij}$, defined as the ratio of the energy stored in each subregion $j$ to the total strain energy stored in the entire beam, can be computed as follows:

$$F_{ij} = \frac{U_{i,j}}{U_i} = \frac{\int_{a_j}^{a_{j+1}} [v_i''(x)]^2 \, dx}{\int_0^L [v_i''(x)]^2 \, dx} \tag{5}$$

Taking into account all identified modes, a damage index $\beta_{ij}$ can then be defined for each subsegment $j$ and mode shape $i$ of the beam as the ratio of the respective fractional energies of the potentially damaged state $d$ and healthy state $u$. As such, the strain energy-based damage indicator $\beta_j$ can be defined as below by considering all identified mode shapes [56].

$$\beta_j = \frac{\sum_{i=1}^{nModes} F_{ij}^d}{\sum_{i=1}^{nModes} F_{ij}^u} \tag{6}$$

*2.3. Modal Flexibility*

Flexibility is defined as the deformation of a structure corresponding to the associated unit load applied at a specific degree of freedom. The flexibility matrix is the inverse of the stiffness matrix, and is generally more straightforward to identify compared to the stiffness matrix. The elements of the flexibility matrix $G_{ij}$ correspond to the displacement at DOF $i$ caused by a unit load applied at DOF $j$. The deformation pattern that a structure attains when a unit load is applied at a specific DOF is provided by the associated column of the flexibility matrix. Damage to a structure that reduces stiffness leads to an increase in the flexibility of the structure. Damage detection using this method is based on computing the modal flexibility matrix for the healthy and potentially damaged states of the structure from the identified mode shapes using the following equation:

$$[G] = \sum_{i=1}^{nModes} \frac{1}{w_i^2} [\phi]_i [\phi]_i^T \tag{7}$$

The change in the flexibility matrix between the two states of the structure can then be calculated as $[\Delta G] = [G_d] - [G_u]$. For each degree of freedom $j$, the change in the flexibility matrix for that degree of freedom is defined as the maximum absolute value of the elements in the $j$th column of the matrix $\Delta G$:

$$max_j = max[\Delta G] = max|\delta G_{ij}| = max|G_{ij}^d - G_{ij}^u| \tag{8}$$

The quantity $max_j$, which is a measure of the change in flexibility between the two states of the structure, can be used to detect and locate the damage. In other words, the column of the $\Delta G$ matrix corresponding to the largest $max_j$ shows the degree of freedom where the damage is located.

*2.4. Modified Modal Flexibility*

An issue related to the modal flexibility method is that $max_j$ represents the maximum of the displacement differences for the two states of the beam. However, for the simply supported beam considered here, the displacements are larger towards the middle of the beam when a unit load is applied at a DOF, and are lower close to the supports. Therefore, the damage index $max_j$ is prone to missing damage near the supports, as the difference between the two small values tends to remain small even for considerable levels of damage near the supports. To alleviate this shortcoming, the damage index $max_j$ calculated using the flexibility method can be normalized using the maximum of the displacements created

by a unit load at DOF $j$. In other words, $max_j$ can be normalized by dividing it to the maximum absolute value of each column of the flexibility matrix; we call this quantity the "modified $max_j$", represented by $max_{mj}$:

$$max_{mj} = max[\Delta G]_j / max|G_j^u|. \tag{9}$$

### 3. Verification Study: Case I

In this case study, the damage-sensitive features are applied on a simply-supported beam represented by a Bernoulli–Euler formulation which represents a hypothetical single-span bridge structure. The modulus of elasticity of the beam and the moment of inertia of its cross-section are 32.7 GPa and 5.47 m$^4$, respectively. In the verification study, it is assumed that the mode shapes acquired from an experimental campaign, where modal identification is normally carried out using accelerations recorded by a set of accelerometers, are identical to those obtained from the eigenvalue analysis of the model. Twelve sensors are located at equal distances of 4.54 m across the span to simulate a conventional instrumentation setup, as depicted in Figure 2a. The hybrid method proposed in this study enables the provision of spatially continuous information at the edges of the bridge. To approximate this approach numerically, the bridge segment between Points 1 and 2 in Figure 2a, i.e., the first segment, is divided into twenty pieces such that the modal displacements are measured at an interval of 0.23 m along this segment (Figure 2b), simulating a case where the vibrations and the associated mode shapes of the first 4.54 m of the beam are extracted using computer vision algorithms [34,35]. Therefore, in the setup of the hybrid monitoring system, 31 sensor locations are taken into account, and more detailed information is obtained near the abutment, where vibration-based damage detection methods conventionally struggle [33]. On the other hand, a conventional sparsely-located sensor setup is used for the rest of the beam, as the damage-sensitive parameters mentioned above have a much higher success rate for detecting the damage located in this region. It should be noted that even denser modal information can be achieved at the edges by using computer vision algorithms and a sensor with higher resolution; however, in this study the total number of segments is limited to twenty for numerical efficiency. In addition, note that only the first half of the beam is considered here for the hybrid monitoring application because of the symmetry of the beam. In a real-life application, both edges of the bridge are assumed to be monitored using computer-based algorithms.

In the presented verification study, the damage is simulated as a 10% reduction in the bending stiffness and is concentrated at a length of 0.45 m near the abutment, as shown by the part highlighted with pink in Figure 2c. The efficacy of the conventional sensor setup (Figure 2a) and the proposed hybrid method (Figure 2b) when used together with the mode shape-based damage parameters is evaluated to detect and locate damage. The change in the structural frequencies due to the simulated damage is negligible, and the first four vertical mode shapes are used to compute the damage-sensitive parameters. The first four modal frequencies are determined as 1.75, 6.91, 14.40, and 26.04 Hz. Note that in reality the identified modal parameters can be altered by environmental factors. Here, we assume that the effects of such factors on modal properties have been eliminated, as this is the common approach in damage detection studies.

Cubic splines are used to increase the spatial resolution of the data when sensors are sparsely located. Here, damage-sensitive parameters are computed using both a conventional and a hybrid setup. As such, the modal displacements and the associated damage-sensitive parameters are calculated at 221 equidistant points (i.e., 220 equally long intervals) for both sensor setups. Figure 3 shows the difference in the mode shapes (MS) obtained using conventional and hybrid methods for the first and third modes. In addition, Figure 3 clearly represents the differences in the estimates of the mode shapes from the conventional and hybrid approaches, which are larger for the third mode and much smaller for the first. This results in an improved estimate of the damage-sensitive parameters, as depicted in Figure 4 and explained further in the following.

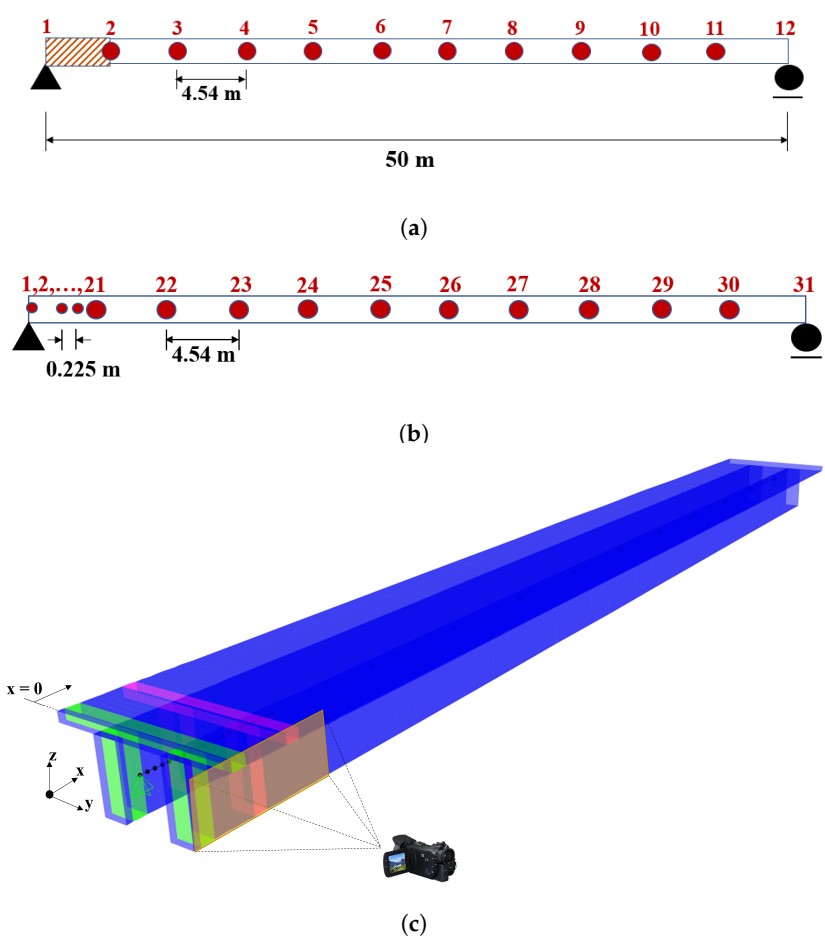

(**a**)

(**b**)

(**c**)

**Figure 2.** Measurement points of (**a**) the conventional monitoring system, (**b**) the hybrid monitoring system, and (**c**) a rendered view of the FE model representing a typical bridge cross-section, the refined first segment, and damage locations.

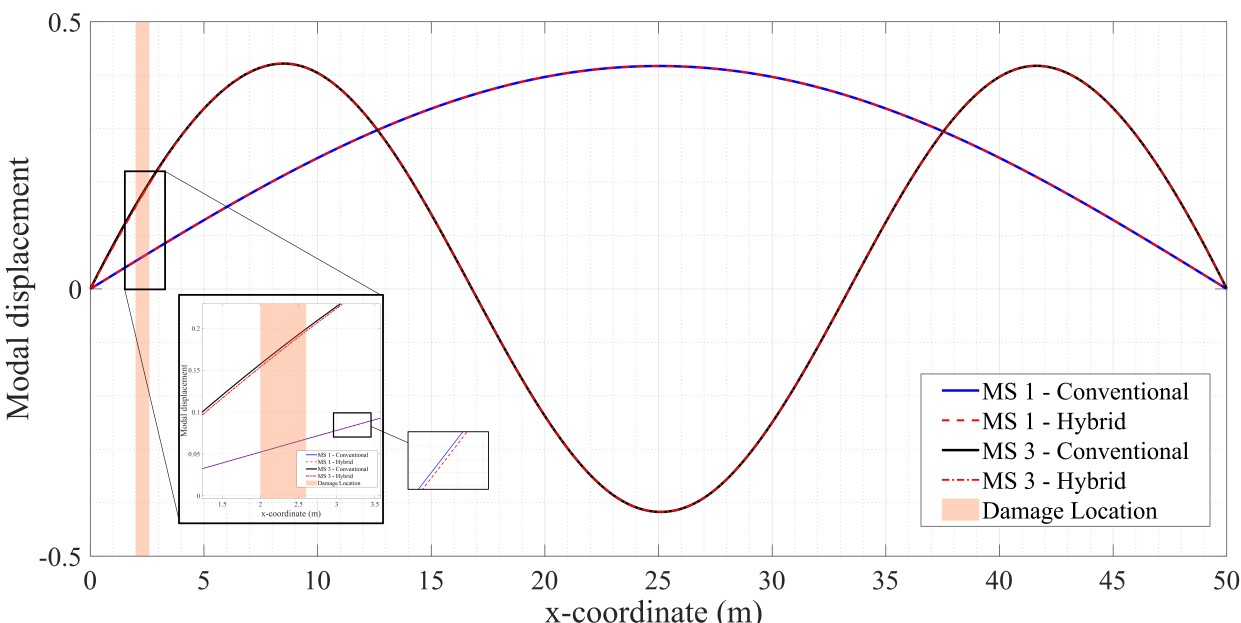

**Figure 3.** Example showing the difference in modal displacements between conventional and hybrid measurements.

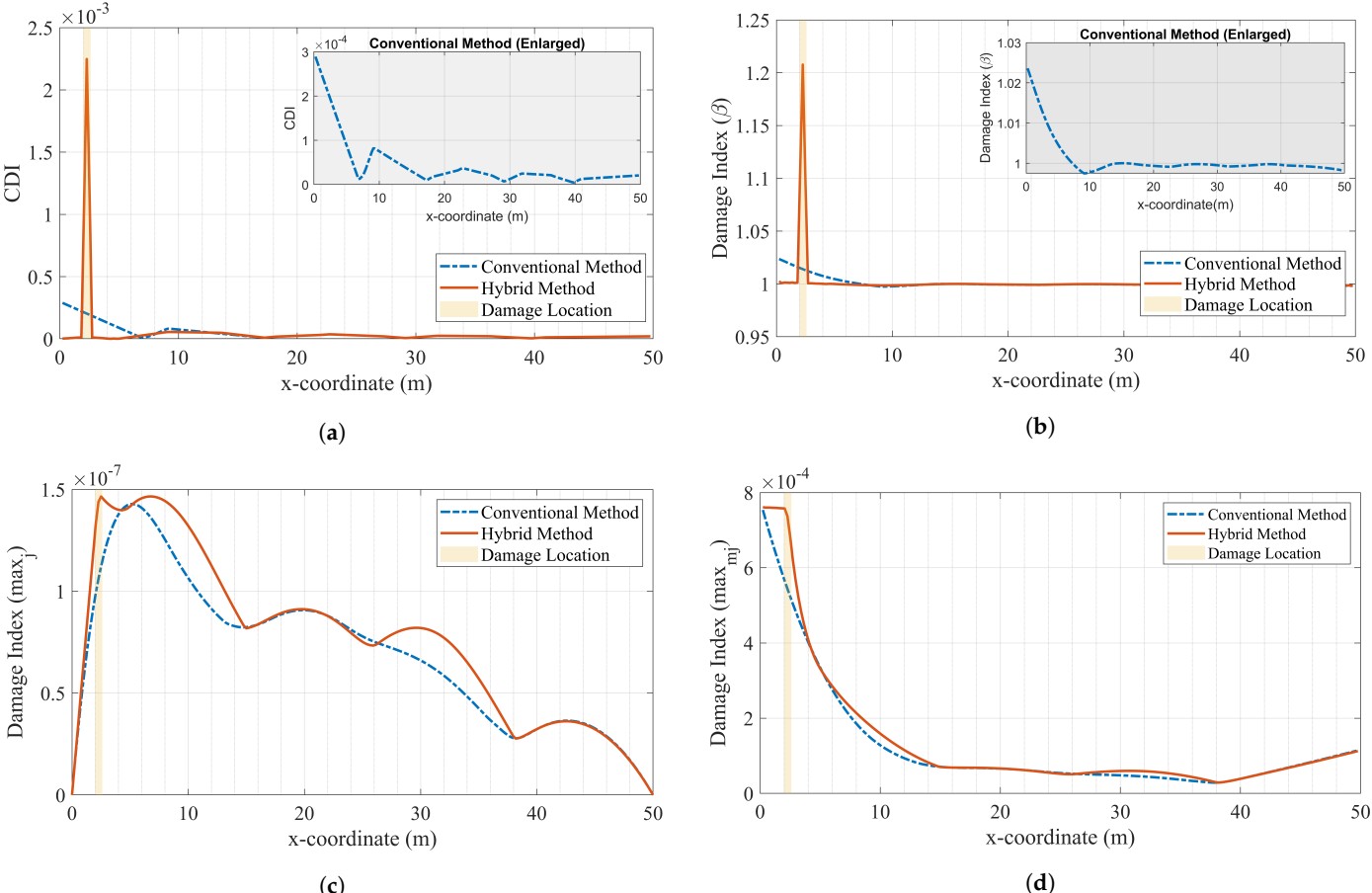

**Figure 4.** Results of damage detection methods for Case I: (**a**) mode shape curvature, (**b**) modal strain energy, (**c**) modal flexibility, and (**d**) modified modal flexibility.

Figure 4 presents the damage-sensitive parameters computed using both the conventional sensor setup and the proposed hybrid methodology. All four damage-sensitive measures provide greatly improved performance when used in conjunction with the proposed hybrid methodology in comparison with the conventional sparse sensor setup. For each of the four damage-sensitive parameters, although damage can be arguably detected using the conventional setup, the smoothing effect of the spline function prevents all four parameters from correctly locating the damage. On the other hand, the damage-sensitive parameters at the damaged location reach much higher values compared to their counterparts at the undamaged locations when the proposed hybrid methodology is used. This difference, which is especially visible for the modal curvature and modal strain energy parameters, shows promise in avoiding the potential negative effects of measurement noise, which is considered in the following section. Furthermore, the location of the damage can clearly be identified with the proposed approach thanks to the increased spatial resolution of the information provided by the computer vision methods close to the abutments.

Finally, Figure 4c clearly demonstrates the inferiority of the flexibility-based damage index compared to the modal curvature and modal strain energy methods. This can be attributed to the relatively low contribution of the higher modes to the flexibility-based damage index, as these have higher modal displacements close to the abutments compared to the first-mode shape. Further, as mentioned above, the flexibility method relies on non-normalized modal displacements that are very small close to the abutments, leading to poor performance in detecting and locating damage at these locations. In fact, normalizing the difference in modal displacements between undamaged and damaged cases with that of the undamaged case, i.e., by using the modified modal flexibility parameter proposed in this

article, provides much improved performance in detecting and locating damage, particularly when used in conjunction with the proposed hybrid monitoring approach (see Figure 4d).

It is intuitive that if the mode shapes present significant changes, then the possibility of the presented damage detection methodologies capturing such changes is increased. Therefore, we inspect the difference between the modal displacements of damaged and healthy cases obtained using only the hybrid monitoring approach in Figure 5. It is clear that the change in the modal displacements is the maximum in the damage interval for all modes. Figure 5 provides significant insight in terms of the effect of the modes considered in the damage detection analysis on the results. One such insight is into the relationship between the location of the damage and the success of damage-sensitive features in detecting it. The choice of the mode shapes to be included in calculating the damage sensitive features affects the results significantly. For example, if the damage is located where the differences in modal displacements are the largest for the fourth mode and this mode is not included in the analysis (as in this example), then the probability of detecting and locating the damage decreases. For simple structures, such as a single-span bridge, ad hoc selection of mode shapes may lead to better results. For more complex structures, one option for weighing the contribution of mode shapes to the results is to multiply their contribution by the modal mass participation ratios corresponding to each mode.

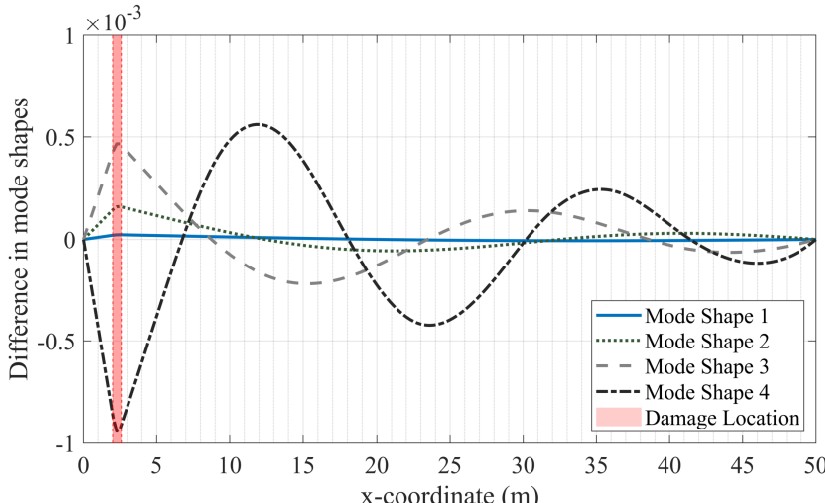

**Figure 5.** Difference in the modal displacements of healthy and damaged states using only hybrid measurements.

## 4. Case Studies

Having verified the feasibility of the proposed approach under ideal conditions in Case I, i.e., where there is no measurement noise, the applicability of the proposed monitoring strategy is investigated through the numerical simulation of measurement campaigns with different sources of excitation. The acceleration response of the bridge under train crossings and ambient vibrations are considered to simulate two common cases of operational modal analysis that aim to identify the mode shapes from measured vibrations. SAP2000 finite element software is used for the numerical simulations conducted in this study. Two cases are simulated: i) the dynamic response of the bridge in damaged and healthy conditions is computed under train crossing effects, and ii) three scenarios of ambient vibration recordings are considered. Various levels of white noise are added to the computed acceleration response at the sensor locations in order to simulate measurement noise. In all cases, the induced damage is assumed to result in a 10% reduction in the bending stiffness of the beam.

### 4.1. Train Crossings

The modal properties of bridges can be obtained experimentally using their acceleration response during train crossings [57–59]. The vehicle–bridge interaction, which affects the dynamic response of both the vehicle and the bridge, can be modeled using various

approaches [60,61]. In this study, the train action is modeled as a series of moving loads that act on the bridge. The ICE-2 train is selected as the train model to represent a modern passenger train. The configuration and axle distances for a typical ICE-2 train are shown in Figure 6. The axle load used in the numerical analysis is 150 kN. Dynamic analyses are performed for a speed of 180 km/h for two localized damage cases. In one case, the damage is located at the same location as the verification study, whereas in the other case it is located even closer to the support. Damage located between 0.23 and 0.68 m from the support (Case II) or between 2.05 and 2.50 m from the support (Case III) are highlighted in green and pink, respectively, in Figure 2c. The damping is modeled using Rayleigh damping anchored at the first and third mode frequencies, and the damping ratio is set to 2%. The duration of the analysis is set to 40 s to ensure that both the forced vibration and free vibration responses of the structure can be captured.

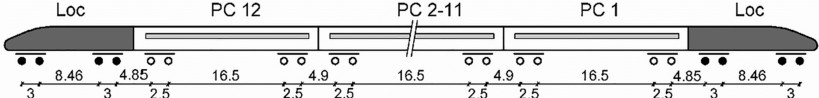

**Figure 6.** Axle distance and configuration for a typical ICE-2 train [62].

Acceleration responses captured at the sensor locations with both the conventional and hybrid approaches are processed using the Frequency Domain Decomposition (FDD) method [63] to estimate the structure's modal frequencies and mode shapes at the discrete sensor locations. The spatial resolution of the mode shapes is increased using cubic spline functions for both the conventional and hybrid sensor setups. Figure 7 shows the recorded acceleration for the undamaged case and the Fourier Amplitude Spectra (FAS) for the sensor data in the hybrid monitoring setup. The first three modal frequencies are identified as 1.75, 6.81, and 14.11 Hz. These three modes are used in damage detection, as the other modal frequencies are above 25 Hz and generally very difficult to capture in real-life measurement campaigns. No additional noise is considered in the vibration records in case of train crossings, as train loads result in high-amplitude vibration, which renders the noise negligible.

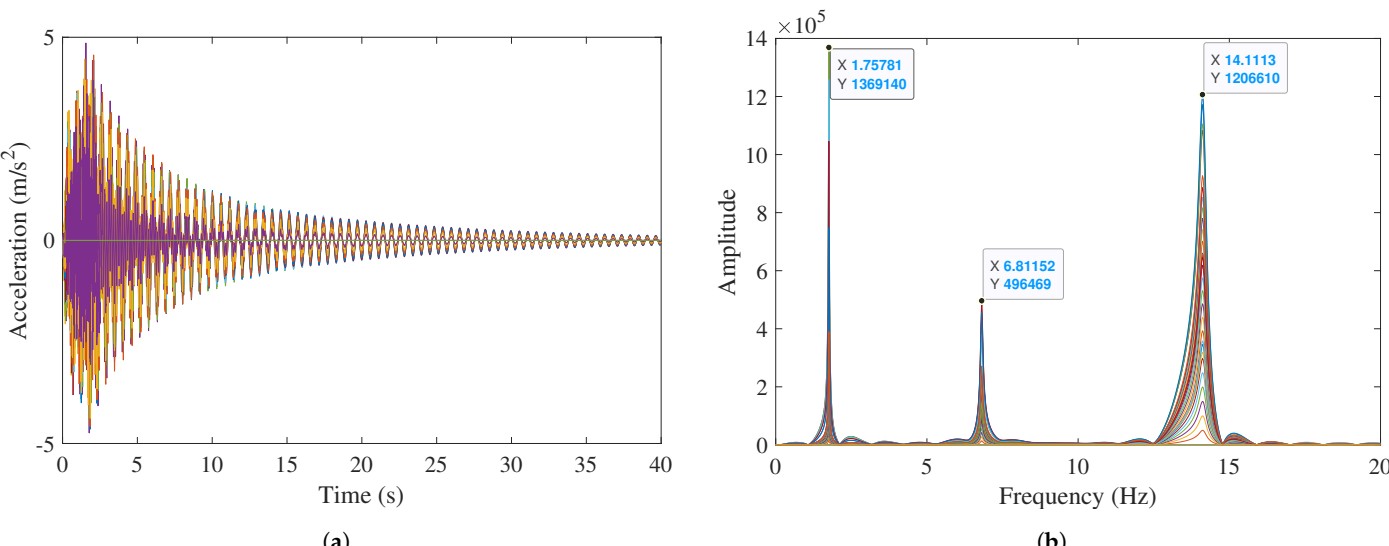

(**a**)　　　　　　　　　　　　　　　　　　　　　　　　　　(**b**)

**Figure 7.** (**a**) Time history of the bridge acceleration responses and (**b**) Fourier amplitude spectra of the vibrations.

### 4.1.1. Damage Next to the Support: Case II

In this case, the damage is located between 0.23 and 0.68 m away from the left support, and is represented as a 10% reduction in the elastic stiffness of the bridge girder. The damage does not produce any noticeable changes in the identified frequencies. The results presented

in Figure 8 show that both the proposed hybrid methodology and the conventional sensor setup can detect damage successfully in this scenario. However, the value of the damage parameter at the location of the damage is much higher for the hybrid approach compared to the undamaged parts of the bridge, leaving no doubt about the presence of damage. On the other hand, for the conventional sensor setup, the value of the damage parameter at the damage location is barely above the values at the undamaged regions of the bridge, and fails to provide a clear picture of the presence or location of the damage.

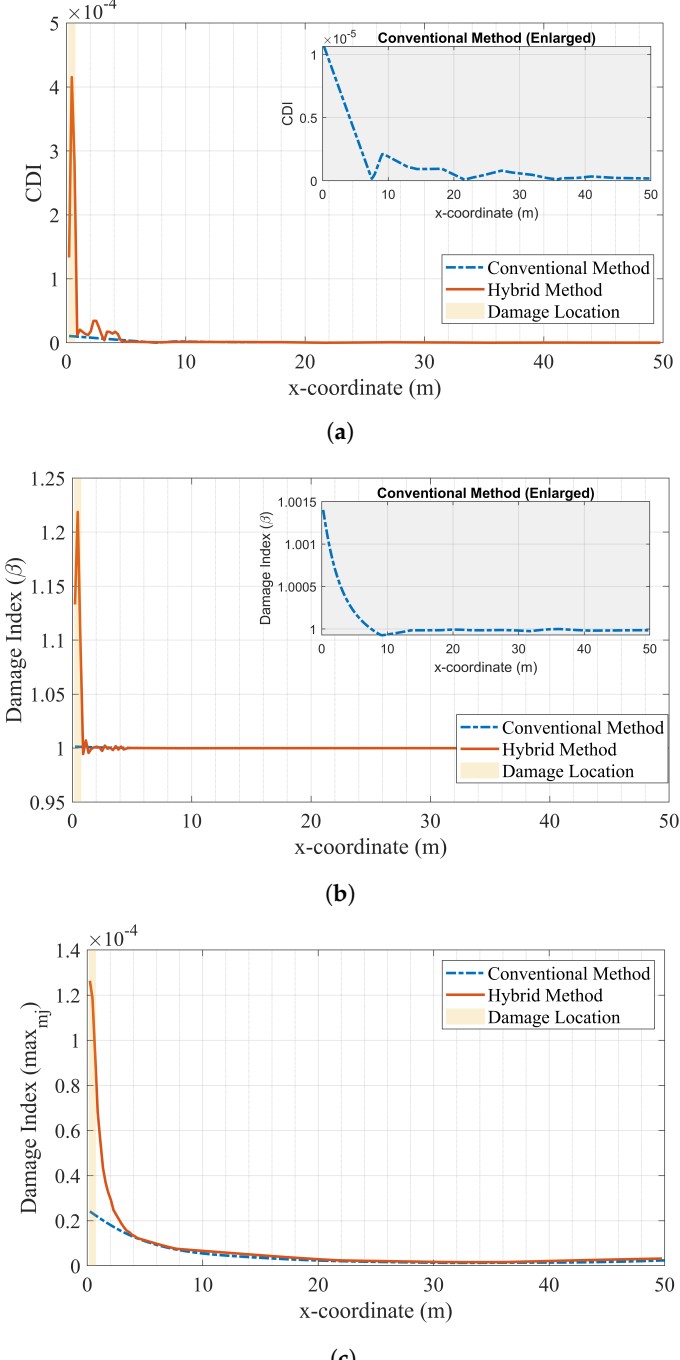

**Figure 8.** Results of damage detection methods for Case II: (**a**) mode shape curvature, (**b**) modal strain energy, and (**c**) modified modal flexibility.

### 4.1.2. Damage near the Support: Case III

In this scenario, the damage is located between 2.05 and 2.50 m from the left support, and is represented as a 10% reduction in the elastic stiffness of the beam elements. The

mode shapes are identified and their resolution increased using the same procedure as in the previous case. The results presented in Figure 9 show that the proposed hybrid procedure provides a very clear picture of damage and its location when used together with the mode shape curvature and modal strain energy parameters. On the other hand, the values of these damage parameters at the damaged location are very close to those at the undamaged locations when the conventional sensor setup is used, creating doubt about the presence of damage. Moreover, the conventional setup cannot accurately capture the damage location correctly, as the damage is located in the middle between two sensors. Instead, it predicts that the damage is at one of the sensor locations.

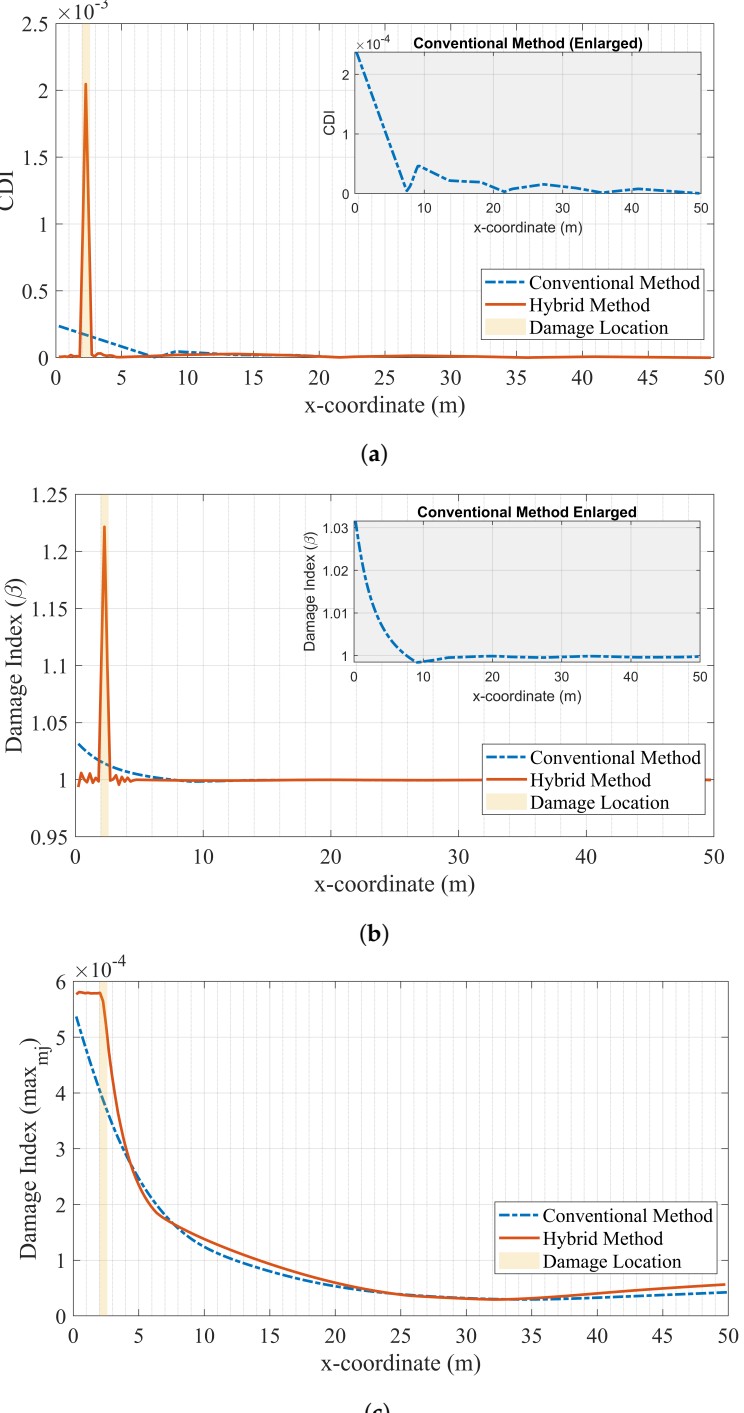

**Figure 9.** Results of damage detection methods for Case III: (**a**) mode shape curvature, (**b**) modal strain energy, and (**c**) modified modal flexibility.

The performance of the conventional and hybrid approaches is much closer to each other when the modified flexibility damage parameter is used. First, the damage parameter values obtained from both approaches are very similar to each other and provide a very clear indication of the presence and location of the damage. However, the location of the damage as estimated by both methods lacks precision, as shown in Figure 9c. When used with the modified flexibility parameter, the conventional approach estimates the damage to be located at the support. The hybrid approach, although it correctly predicts the damage location, overestimates the length of the damaged portion.

### 4.2. Ambient Vibration

The OMA of bridges under ambient vibrations is arguably the most widely used method for structural identification and damage detection of bridges. The response of the structure to the ambient vibrations caused by a variety of actions such as wind and nearby traffic is measured and processed to obtain the modal properties. In most of the system identification methods used in OMA, the input is unknown and assumed to be stationary white noise. To simulate such applications, Gaussian white noise acceleration with a total duration of six hundred seconds is applied to the supports of the structure in the vertical direction and the response is measured at the sensor locations with a sampling frequency of 100 Hz. Figure 10 displays the bridge acceleration responses that are significantly lower compared to the response during the train crossing (Figure 7a).

The response data are then processed using the FDD method to obtain the power spectral density (PSD) graph presented in Figure 10 for the healthy bridge. Peaks corresponding to the first and third modal frequencies are identified as 1.75 and 14.11 Hz from the PSD, and the associated mode shapes are determined. Note that the second modal frequency is not visible in the PSD, as the modal mass participation of the second mode is zero due to the perfect symmetry of the beam; thus, the second mode does not contribute to the numerical response. Therefore, only the first and third modes are used to calculate damage-sensitive features when the ambient vibration response is used. For all cases in this section, the damage is located between 2.05 and 2.50 m away from the left support and is represented as a 10% reduction in the elastic stiffness of the beam elements.

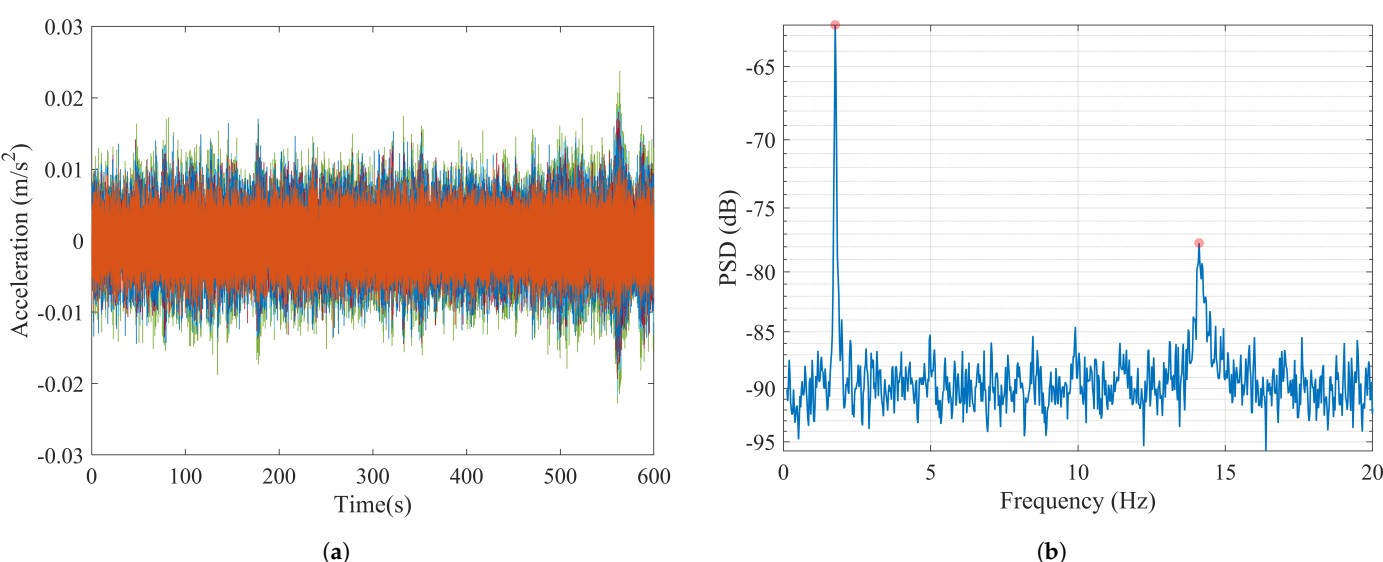

**Figure 10.** (**a**) Ambient vibration response measured at the sensor locations and (**b**) PSD graph of vibration responses for Case VI-healthy bridge.

In OMA with ambient vibrations, noise levels are of great concern for the accuracy of system identification algorithms due to the very low levels of vibration response of the bridge, which accentuates the measurement noise. To investigate the sensitivity of the proposed methodology to noise and compare its performance with conventional sensor

setups in detecting and locating damage, the noise-free acceleration signal at each sensor location is polluted with Gaussian noise for both the conventional setup and the hybrid approach:

$$\ddot{u} = \ddot{u}_0 + \beta\gamma(max(abs(\ddot{u}_0)))$$
(10)

where $\ddot{u}_0$ is the noise-free acceleration record, $\ddot{u}$ is the noisy acceleration signal, $\beta$ is a parameter that determines the level of noise, and $\gamma$ is a random number with a standard normal distribution. Three cases, with the signals are contaminated by (i) 2% noise, (ii) 15% noise, and (iii) 30% noise, are considered here in order to account for low, moderate, and high levels of noise, respectively. The signal-to-noise ratios (SNR) can be computed as follows: (i) SNR=34 dB for 2% noise, (ii) SNR=16.5 dB for 15% noise, and (iii) SNR=10.5 dB for 30% noise, using

$$SNR = 20 \, log_{10}(\frac{1}{\beta}),$$
(11)

where $\beta$ is the noise level.

### 4.2.1. Low Noise: Case IV

In the case of low measurement noise, the results presented in Figure 11 demonstrate that this level of noise does not impact the performance of the proposed hybrid method in detecting and locating the damage when the mode shape curvature and modal strain energy methods are used. As in the case of no noise (Figure 4), the hybrid approach can clearly detect and locate the damage, while the conventional approach fails to locate the damage and provides a much smaller damage index than the proposed hybrid approach. On the other hand, the proposed normalized modal flexibility damage index is much more prone to measurement noise, as the performance of both hybrid and conventional approaches in locating damage declines significantly even for the low level of noise applied. It can be stated with confidence that the addition of further noise would further reduce the accuracy of the normalized modal flexibility parameter; therefore, this parameter is not included in the discussion of higher levels of noise.

### 4.2.2. Moderate Noise: Case V

When the noise level increases from low (2%) to moderate (15%), the performance of the conventional approach in detecting and locating damage decreases significantly for the modal curvature damage index, as shown in Figure 12a. In addition to the left support, which is close to the actual damage location, the damage index attains non-zero values at the mid-span and close to 40 m from the left support, leading to uncertainty in damage detection. Further, it renders correct location of the damage impossible. In addition, the conventional method fails to correctly locate the damage when using the modal strain energy parameter, as seen in Figure 12b. On the other hand, Figure 12 shows that the hybrid approach can clearly detect the damage and locate it even with moderate noise levels by using both the modal curvature and modal strain energy damage parameters, as the hybrid methodology remains unaffected by the moderate noise level.

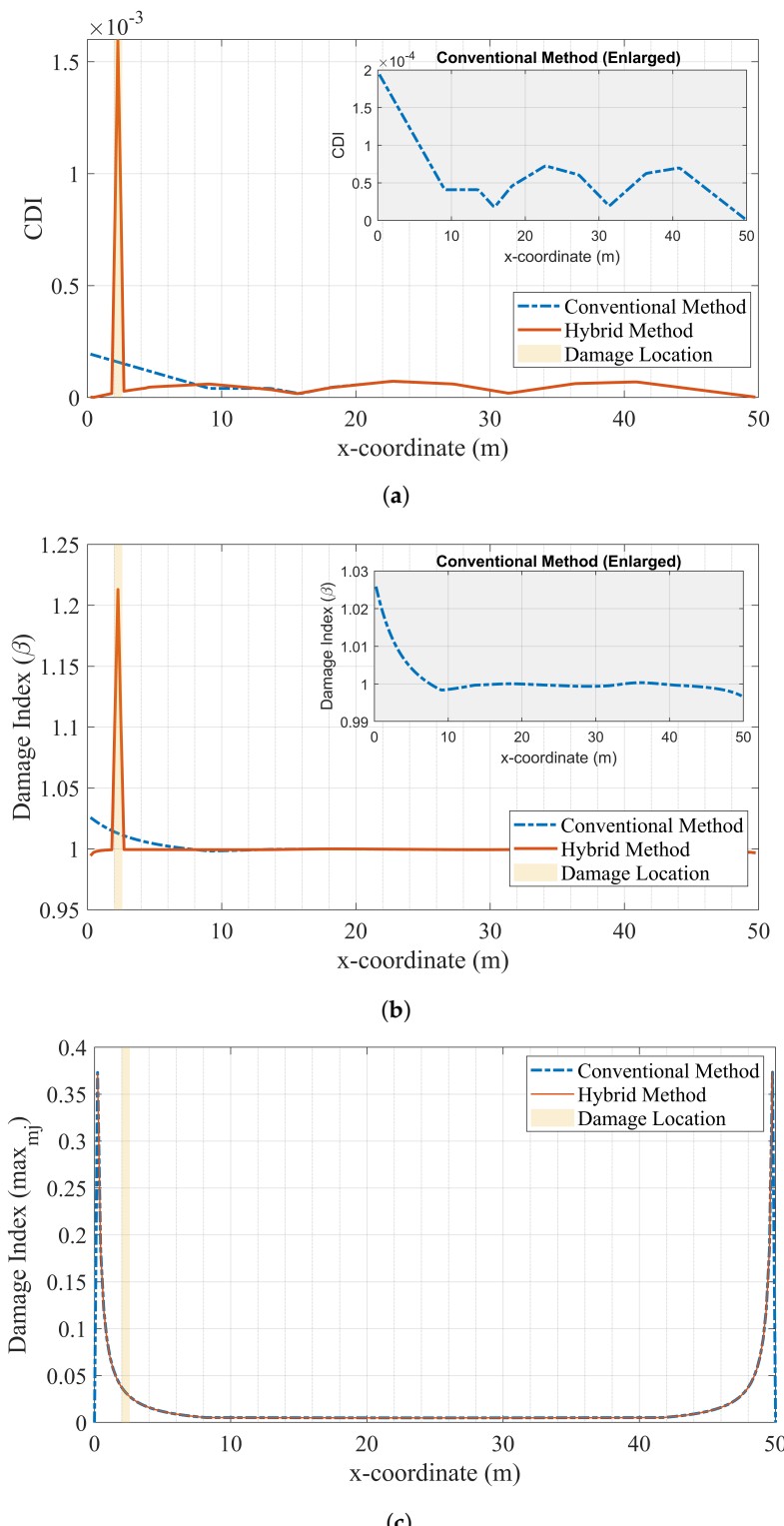

**Figure 11.** Results of damage detection methods for Case IV (low noise): (**a**) mode shape curvature, (**b**) modal strain energy, and (**c**) modified modal flexibility.

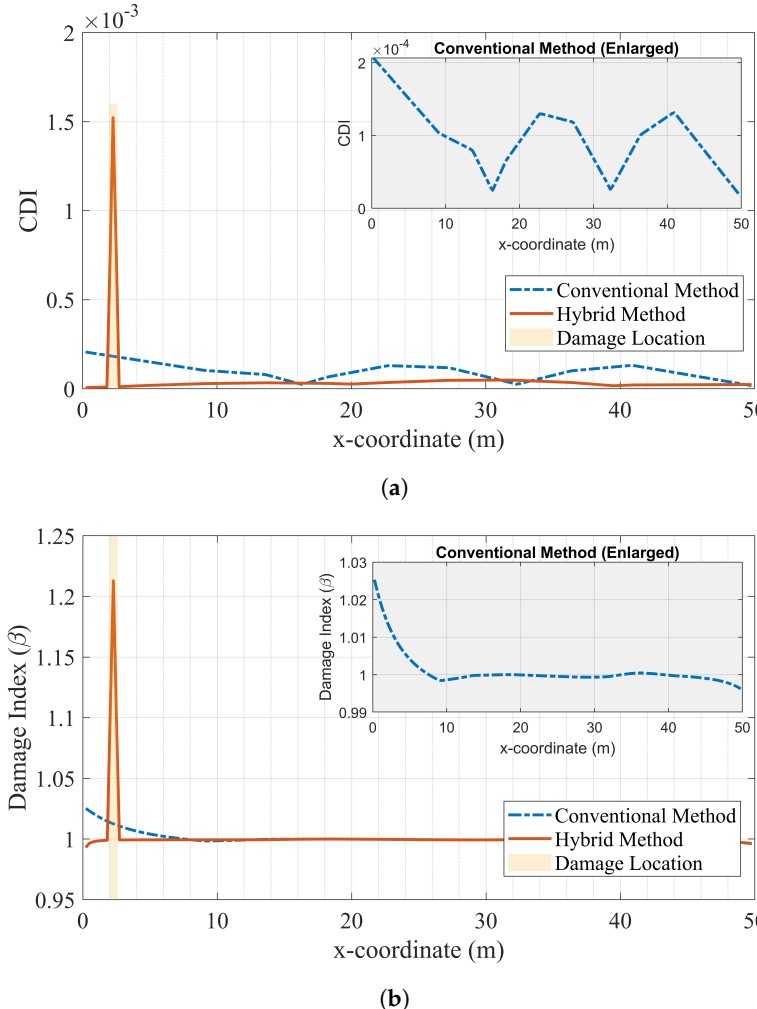

**(a)**

**(b)**

**Figure 12.** Results of damage detection methods for Case V (moderate noise): (**a**) mode shape curvature and (**b**) modal strain energy.

### 4.2.3. High Noise: Case VI

Figure 13 shows that the proposed hybrid methodology is able to clearly detect and locate the damage even with high levels of measurement noise when used with either the modal curvature or modal strain energy damage parameters. Recalling that the simulated damage is minor, as it only leads to a 10% decrease in the flexural stiffness of a 0.45 m strip of a 50 m long bridge, and the added level of noise is very high, as indicated by the signal-noise-ratio of 10.5 dB, it can be stated that the proposed hybrid approach is very robust against measurement noise even when detecting and locating low levels of damage. However, the conventional sensor setup when used together with the modal curvature damage index provides a very confusing picture and fails to detect and locate the damage. Although the performance of the conventional approach increases when used with the modal strain energy parameter, it remains inferior to the performance of the hybrid approach and cannot locate the damage, as shown in Figure 13b.

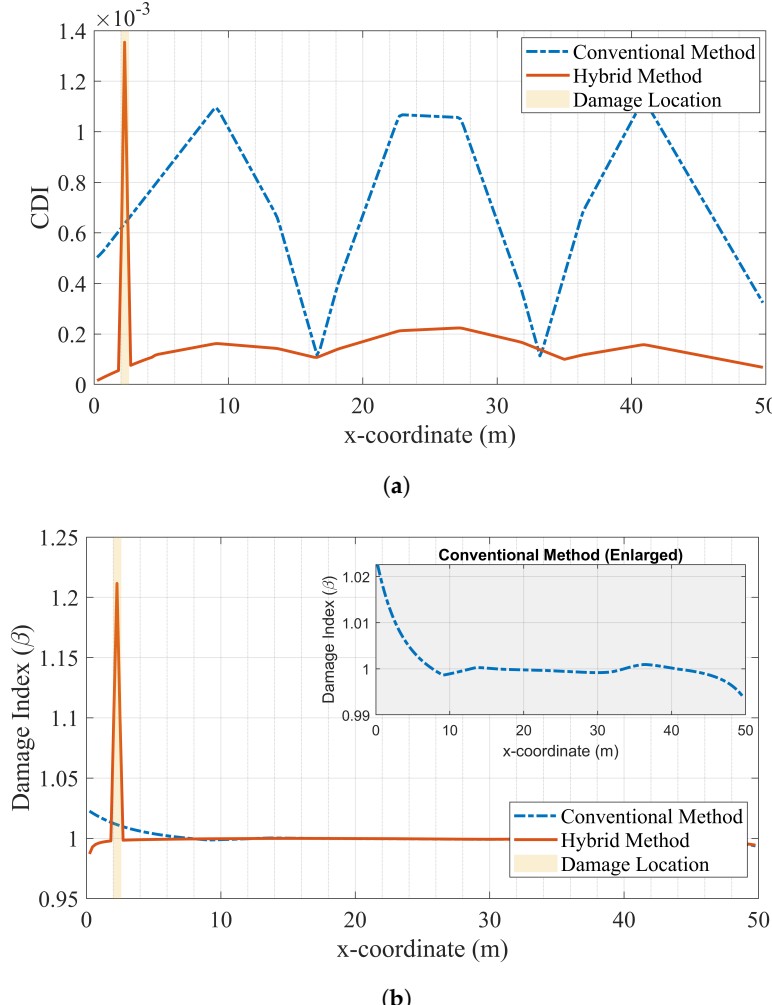

**Figure 13.** Results of damage detection methods for Case VI (high noise): (**a**) mode shape curvature and (**b**) modal strain energy.

## 5. Conclusions

In this paper, we have proposed a hybrid monitoring approach based on data fusion from conventional acceleration sensors and computer vision to detect and locate damage in bridges. The efficacy of the proposed method compared to conventional sparse sensor setups was investigated through numerical analyses of a 50-m-long simply-supported bridge. The bridge was excited by two vibration sources of significantly different levels: a passenger train crossing and ambient vibrations. The mode shapes of the bridge in damaged and healthy states were identified from the data at the discrete sensor locations using the FDD algorithm. It was assumed that the sensor used to simulate computer vision provided vibration measurements at 21 discrete points with a distance of 0.23 m between consecutive points, significantly increasing the spatial resolution of the data. Four damage parameters were used to identify two damage scenarios simulated by reducing the flexural stiffness of a 0.45-m strip of the bridge by 10%. Three different levels of measurement noise were added to the ambient vibration measurements in order to investigate the robustness of the proposed method to measurement noise.

The results indicate that the use of cameras at the ends of the bridge in conjunction with traditional sensors in the middle of the bridge provides a very attractive alternative for vibration-based structural health monitoring applications. Computer vision was able to provide detailed and continuous information in the sensitive regions of the investigated beam, and improved the mode shape estimates in comparison to those obtained from the conventional health monitoring method using discrete sensor locations. When using the

hybrid approach in which vibration data were gathered from computer vision sensors at the ends of the bridge, the performance of both the mode shape curvature and modal strain energy damage parameters increased significantly. On the other hand, these parameters failed to detect and locate damage when based on a conventional sparse sensor setup for damage close to the abutments. For all of the considered damage scenarios, excitation sources, and noise levels, the hybrid methodology was able to clearly detect and locate the damage using the modal curvature and modal strain energy parameters. On the other hand, the same parameters provided much inferior estimates when used together with the conventional sparse sensor setup. The proposed hybrid approach was proven to be very robust against measurement noise, as the damage could be detected and located clearly even with very high levels of measurement noise.

This study demonstrates the applicability and strong potential of the hybrid monitoring system for the detection of bridge damage. A simply-supported bridge was chosen for its ubiquitous presence, as it is arguably the most commonly observed and studied type of bridge. Therefore, the hybrid monitoring approach was first tested on this bridge typology. It should be noted that bridges can be much more complex, with several spans and different boundary conditions, and the efficacy of the proposed method must be tested on such complex bridges in future work. In addition to the assumptions and limitations stated throughout the article, adequate consideration should be given to the uncertainties related to the damage detection problem. They include, but are not restricted to, the modeling of the investigated structure and its boundary conditions [64], variations in material properties and loading [65], and uncertainties in mode shape estimations [66]. Future research will focus on the practical aspects of fusing the modal data obtained from heterogeneous sources, i.e., the camera and conventional accelerometers. In the practical application of the hybrid method, an acceleration sensor could be placed in the middle of the area monitored with cameras and used to cross-validate the proposed method and conventional monitoring by generating a common measurement point. Future studies will also include practical applications in the laboratory and on existing bridges.

**Author Contributions:** Conceptualization, S.G. and E.E.; methodology, S.G. and E.E.; software, S.G.; formal analysis, S.G.; writing—original draft preparation, S.G.; writing—review and editing, E.E.; visualization, S.G. All authors have read and agreed to the published version of the manuscript.

**Funding:** This research received no external funding.

**Institutional Review Board Statement:** Not applicable.

**Informed Consent Statement:** Not applicable.

**Conflicts of Interest:** The authors declare no conflict of interest.

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
