# Peer review of "A Hybrid Method for Vibration-Based Bridge Damage Detection"

_remotesensing, doi:10.3390/rs14236054_

Round 1

Reviewer 1 Report

The authors proposed a hybrid health monitoring methodology based on the fusion of data from conventional accelerometers and computer vision-based measurements. The proposed method is novel. I identified, however, the following open issues:

1. The research gap is not explicitly identified. More related work of Structural Health Monitoring methods based on computer vision should be introduced to reflect the novelty of the work.

2. Section 2 introduces several vibration-based damage indicators. They have little relevance to the innovation of this paper and are recommended to be streamlined or placed in the appendix.

3. In Section 3. The authors proposed to use cameras at the bridge ends and sensors at the middle of the bridge for damage detection. Why not add a sensor at the end of the bridge (for example, the middle of the first 4.54m) at the same time for cross-validation.

4. In Section 4.1. There should be a separate subsection for case I.

5. Figure 8 is not referenced throughout the paper.

6. Phrases only need to give the full expression on the first occurrence. such as OMA, DOF

7. The authors should check the language and grammar carefully.

Author Response

Thank you for your comments. Please see the attached word document.

Reviewer 2 Report

The manuscript mainly develops a method to detect bridge damage, combining the use of conventional accelerometers (in other parts of the bridge) with computer vision (near the abutments). To improve the manuscript, the authors are encouraged to appropriately address/answer the following comments/questions.

1. The proposed method is investigated based on numerical examples for a simply-supported bridge. To the reviewer's best knowledge, real-world bridges can be more complex, for instance, with different boundary conditions. Is it possible that the authors explain in detail why the simply-supported bridge is enough for the validation?

2. Have the changes in modal properties due to environmental factors, e.g. temperature being considered? Modal parameters e.g. natural frequency and damping ratio can be time-variant.

3. Furthermore, the study considers various levels of white noise. Do the authors consider other types of noises as actually existing in the real-world data?

Probably, the-state-of-the-art of the research area is not yet considered those aspects as well? However, it can be inspired by others. At least, in the future work part (after the conclusion), some relevant aspects can be mentioned/discussed. Correspondingly, some papers can be referred/discussed, particularly (if any) those already published in the manuscript submission journal in relevant fields.

Of course, it is notable that acceptance of the manuscript is not conditional upon the citations/discussions of the recommended papers, e.g.:  

[1] https://doi.org/10.1016/j.ymssp.2019.03.021 (structure geometry uncertainties mentioned and considered in this paper)

[2] https://doi.org/10.1016/j.ymssp.2021.108557 (modal properties uncertainties mentioned and considered in this paper)

The authors can find more relevant papers.

Author Response

(The authors gave the same response as above.)
